# Locality-Sensitive Hashing for Efficient Hard Negative Sampling in Contrastive Learning

## Abstract

Contrastive learning is a representational learning paradigm in which a neural network maps data elements to feature vectors. It improves the feature space by forming lots with an anchor and examples that are either positive or negative based on class similarity. Hard negative examples, which are close to the anchor in the feature space but from a different class, improve learning performance. Finding such examples of high quality efficiently in large, high-dimensional datasets is computationally challenging. In this paper, we propose a GPU-friendly LSH scheme that quantizes real-valued feature vectors into binary representations for approximate nearest neighbor search. We demonstrate on several datasets from both textual and visual modalities that our approach outperforms other hard negative mining strategies in terms of computational efficiency without significant performance degradation.

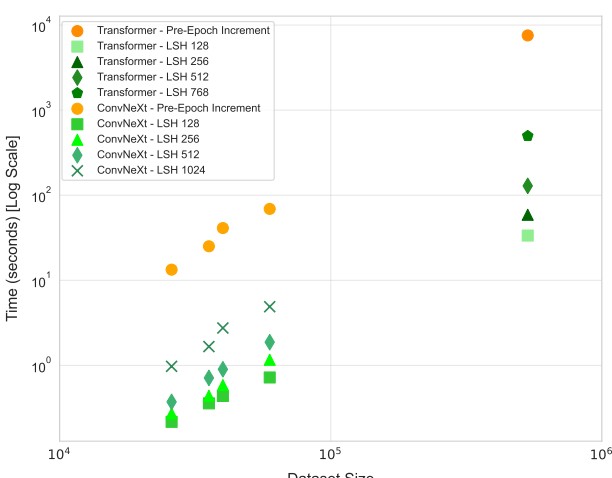

Figure 1: We compare search time relative to dataset size, showing results with ConvNeXt and Transformer models. Both use LSH-based feature encoding with varying bit sizes, along with pre-epoch HN sampling using float32 embeddings and their respective model output sizes.

## 1. Introduction

Contrastive learning builds on the principle of distinguishing positive (similar) examples from negative (dissimilar) examples, and aims to learn a representation space in which similar data points are closer together than dissimilar ones. Unlike supervised classification, which relies on hard label-defined boundaries, contrastive learning provides a learning strategy for tasks where such strict boundaries are inadequate. Scenarios for contrastive learning vary from person re-identification (Hermans et al., 2017) or face verification (Schroff et al., 2015), player re-identification (Zhang et al., 2020; Habel et al., 2022), up to cross-view geo-localization (Deuser et al., 2023a; Zhu et al., 2022; Deuser et al., 2024; 2023b), sentence and text retrieval (Reimers & Gurevych, 2019; 2020), multi-modal retrieval (Radford

et al., 2021; Zhai et al., 2023), and product search (Patel et al., 2022; An et al., 2023). These tasks exemplify the success and versatility of contrastive learning across diverse domains. An example of the need for contrastive learning in these scenarios is product search. Items of clothing may look very similar, almost identical, but belong to different categories, such as a sweater compared to a sweatshirt. In text retrieval, where sentences with different structures and vocabularies can convey the same meaning, the challenge is even greater. These use cases underscore the need for embeddings that capture nuanced similarities without enforcing rigid class separations.

Batch composition sampling strategies are crucial in contrastive learning, as they significantly impact training effectiveness (Wu et al., 2017). Research has shown that incorporating negative examples close to the anchor sample, called Hard Negative (HN), can improve learning outcomes (Wu et al., 2017; Galanopoulos & Mezaris, 2021; Wang et al., 2019; Yuan et al., 2017; Hermans et al., 2017; Cakir et al., 2019; Xuan et al., 2020). However, with modern datasets

[1]Anonymous Institution, Anonymous City, Anonymous Region, Anonymous Country. Correspondence to: Anonymous Author <anon.email@domain.com>.

Preliminary work. Under review by the International Conference on Machine Learning (ICML). Do not distribute.

containing millions (Radford et al., 2021) to billions (Jia et al., 2021) of samples, computing and training on all possible negative combinations is impractical. Pre-extracted HNs, though often more effective than random sampling, do not adapt to changes in the embedding space during training. This limits their effectiveness. Therefore, an efficient dynamic selection of HNs based on specific criteria is crucial to maximize training effectiveness.

A common strategy for HNs calculation (Wang et al., 2019; Yuan et al., 2017; Hermans et al., 2017; Cakir et al., 2019; Xuan et al., 2020) is the within-batch selection, based on a pre-defined criteria. The within-batch calculation is computationally effective as the HNs are selected dynamically during training. However, pre-epoch HNs sampling, which computes negatives globally, offers the advantage of a more comprehensive view of the dataset leading to higher performance (Deuser et al., 2023a). Unfortunately, this introduces significant computational complexity, making it infeasible for large datasets, as shown in Figure 1.

To address computational inefficiency of pre-epoch HN sampling, we propose a lightweight Approximated Nearest Neighbor (ANN) approach that leverages Locality-Sensitive Hashing (LSH) (Charikar, 2002) to reduce search time and space costs. We store and retrieve HNs efficiently by encoding approximate embeddings in a compact binary space, enabling fast queries while maintaining a global view of the dataset. This accelerates pre-epoch sampling, boosting training efficiency without sacrificing effectiveness.

Our work first explores HN sampling methods and introduces LSH as an ANN approach. Using this foundation, we design a training process and evaluate it on six datasets spanning two modalities. We then compare HN-based performance gains against random negatives and those mined with our LSH-based method. Finally, we analyze the hardness and relevance of the mined HN in relation to real HN identified by cosine similarity.

To summarize, we contribute:

- A lightweight and efficient framework for HN sampling using LSH, offering a global view of the dataset while keeping computational costs low during training.

- A comprehensive analysis of LSH on six datasets in the context of supervised contrastive learning, demonstrating its effectiveness on dynamic embeddings during training on multiple datasets from two modalities.

- A demonstration of our method drastically reducing the complexity of training time and providing an efficient and scalable HN-sampling strategy without severe performance degradation despite its approximate nature.

## 2. Related Work

Existing mining strategies for contrastive learning fall into two categories: within-batch sampling and pre-epoch sampling. We briefly review both strategies.

### 2.1. With-In Batch Sampling

Simo-Serra et al. refined within-batch sampling by selecting HNs based on loss values computed after the forward step (Simo-Serra et al., 2015). Samples are chosen randomly at the start of each epoch, with backward gradients computed only for high-loss cases. Similarly, triplet loss (Schroff et al., 2015) enhances HN sampling. Schroff et al. employed it in an online mining scheme, selecting HNs within a batch using $\ell_2$ distance.

Subsequent work (Wu et al., 2017) introduces semi-HN sampling, as mining only the hardest examples can cause model collapse. Others (Hermans et al., 2017) compared multiple mining strategies for triplet loss in person re-identification, showing that selecting the hardest positive and negative within a batch outperforms prior work (Oh Song et al., 2016; Ding et al., 2015). Hermans et al. investigated offline hard mining as well (Hermans et al., 2017). However, selecting the hardest samples across the entire dataset led to suboptimal performance, causing model collapse with standard triplet loss and hindering training.

Yuan et al. proposed a cascaded model to identify HNs at different network stages (Yuan et al., 2017), enabling the model to focus on hard examples when they are the most difficult to distinguish, improving learning effectiveness.

Another strategy is mining informative pairs by comparing negative pairs with the hardest positive pairs and vice versa (Wang et al., 2019). Wang et al. further refined this mining strategy with a soft weighting scheme to more accurately prioritize the selected pairs (Wang et al., 2019). For positive sampling Xuan et al. found out that easiest samples can provide higher generalization as the embedding maintains intra-class variance (Xuan et al., 2020).

### 2.2. Pre-Epoch Sampling

While previous work focusses on in-batch strategies, Cakir et al. take a different approach by defining the batch composition during the training epoch (Cakir et al., 2019). They use WordNet (Pedersen et al., 2004) similarities between classes to determine which classes should be sampled together, effectively introducing harder-to-differentiate samples. In a previously mentioned study, Hermans et al. also explore offline HN mining (Hermans et al., 2017), but in line with the results of (Wu et al., 2017), they found that HNs can lead to model collapse with standard triplet loss.

While most of the previous work focused on image retrieval

Gillick et al. introduced HN sampling for entity retrieval in Natural Language Processing (NLP) (Gillick et al., 2019). They encode all mentions and entities, identifying the 10 most similar entities after each epoch. If an incorrect entity ranks higher than the correct one, it is treated as an HN. Qu et al. enhance this by adding a cross-encoder denoising mechanism to reduce false negatives (Qu et al., 2020).

Xiong et al. pioneer the use of ANN by storing embeddings in a database during training and performing sampling on asynchronously updated indices (Xiong et al., 2020). In cross-view geo-localization, Deuser et al. highlight the significant performance improvements achievable with HN sampling (Deuser et al., 2023a), and show that the InfoNCE loss (Oord et al., 2018) avoids the collapsing problems often associated with triplet loss. However, their method incurs significant computational and storage costs due to the need to compute the entire similarity matrix.

### 2.3. Research Gap

Xiong et al. construct an index using full vector embeddings and asynchronously update the embeddings for the entire dataset every few batches, significantly increasing computational overhead and double the GPU resource requirements (Xiong et al., 2020). In contrast, we investigate whether a lower-dimensional binary representation is sufficient to retrieve high-quality HNs.

## 3. Method

### 3.1. Preliminary

The primary goal of contrastive learning is to bring positive pairs closer together in the embedding space while pushing negative pairs farther apart. In a supervised setting, positive pairs consist of samples with the same label, while negative pairs have different labels. Previous work (Deuser et al., 2023a; Xiong et al., 2020; Wang et al., 2019; Xuan et al., 2020; Cakir et al., 2019) has shown that the selection of negative samples can significantly affect the learning process, either by speeding it up or by improving generalization. In this work we first want to analyze the theoretical properties of our approach based on ANN for the selection of HNs.

We aim to find a suitable embedding $Y \subset \mathbb{R}^d$ in a d-dimensional real vector space for some input data $X$ that is parametrized by a neural network, i.e. $f_\theta : X \to Y$. To do so, we first want to establish the InfoNCE loss on the embedding space, a contrastive loss function defined by (He et al., 2020) as

$$\mathcal{L}_c(y_1, \ldots, y_K) = -\log \frac{\exp\left(\frac{c^\top y_+}{\|c\|\|y_+\|}/\tau\right)}{\sum_{i=1}^{K} \exp\left(\frac{c^\top y_i}{\|c\|\|y_i\|}/\tau\right)}$$

on a batch of size $K$. $c$ is called the anchor point, $y_+ \in$

$y_1, \ldots, y_K$ serves as its positive sample from the identical class and other $y_i$ serve as its negative samples from arbitrary different classes. $\tau$ serves as a temperature parameter, controlling how concentrated the features are in the representation space. If the cosine similarity between the anchor and its positive sample, defined as

$$\text{sim}(c, y_+) = \frac{c^\top y_+}{\|c\|\|y_+\|}$$

is high, the loss decreases. Vice versa, if the similarity between the anchor and its negative samples is high, the loss increases.

During training, we iteratively sample anchors and the corresponding batches of their positive and negative samples, to calculate the derivative of the loss w.r.t. to $\theta$, to achieve a better embedding, pushing $c$ closer to its positive sample $y_+$ and further away from its negative samples. $\theta$ denotes the weights of the neural network.

According to (Schroff et al., 2015), a crucial step is choosing meaningful positives and negative samples for the anchor to achieve fast convergence, i.e. the similarity between anchor and positive sample is lower than between anchor and negative samples:

$$\text{sim}(c, y_+) \leq \text{sim}(c, y_i) \,\forall i$$

Given an anchor $c$, we call

$$y_- = \arg \max_{y \in Y : y \neq y_+} \text{sim}(c, y)$$

its HN sample. It is the most similar instance of another class to the anchor. Thus, the batch must include the $K - 1$ hardest negatives.

Since every element in the dataset can be an anchor with its corresponding batch of HNs, traditional pre-epoch sampling calculates the full similarity matrix $S$. Each entry $S_{ij} = \text{sim}(y_i, y_j)$ represents the cosine similarity between the pairs of embeddings. The calculation of this matrix is computational expensive and memory-intensive due to its size which scales quadratically to dataset size $M$.

To mitigate these computational expensive and memory-intensive drawbacks, we employ an ANN method (Har-Peled et al., 2012; Charikar, 2002) resulting in reduced computational complexity and time. Our approach to ANN utilizes LSH, which we will elaborate on in the following.

### 3.2. Locality Sensitive Hashing

During training, it is essential to query stored vectors to identify HNs after each epoch. However, storing all these vectors can become space intensive, especially as the number of data points grow. To address this challenge, we

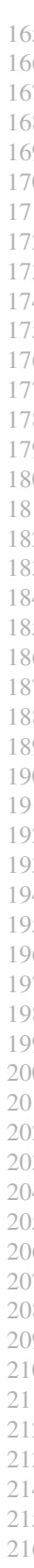

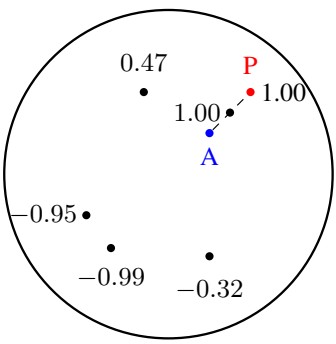 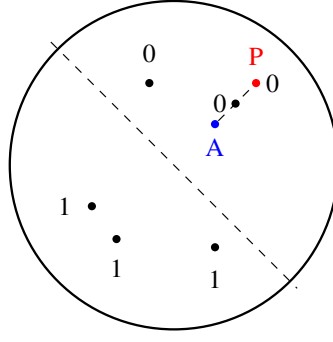 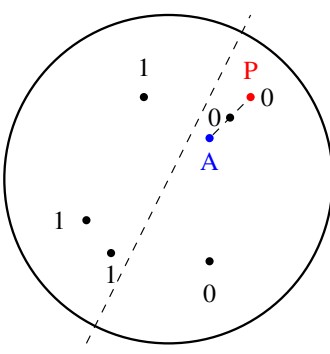

No Hyperplane (Cosine Similarities)     Random Hyperplane $\frac{1}{\sqrt{2}}(1,1)$     Random Hyperplane $\frac{1}{\sqrt{5}}(2,-1)$

Figure 2: Illustration of the anchor (A, blue), positive (P, red), and several negatives (N, black). **Left:** The raw cosine similarities between the anchor and the negatives are shown, which are commonly used to determine HNs (examples that are very close to the anchor). **Middle and Right:** Two examples of randomly sampled hyperplanes ($\frac{1}{\sqrt{2}}(1,1)$ and $\frac{1}{\sqrt{5}}(2,-1)$) are provided, demonstrating how the HNs have a high probability of being mapped to the same side of the hyperplane as the anchor. The Hamming distance is defined as the number of hyperplanes separating the embeddings. Thus, higher cosine similarity corresponds to a smaller Hamming distance, enabling effective identification of HNs.

adopt a binarization approach inspired by LSH, which significantly reduces storage requirements while maintaining the ability to efficiently retrieve ANNs. Following previous work (Har-Peled et al., 2012; Datar et al., 2004; Andoni et al., 2015), we implement this approach by sampling a random rotation (i.e. the vectors are othonormal) matrix.

$$R \in \mathbb{R}^{b \times d}$$

where $d$ denotes the dimensionality of the embedded feature vector $y \in Y$, and $b$ specifies the bit dimension of the encoded feature vector. The embedded dataset $Y$ is first transformed using the random matrix $R$ and then in every dimension centered around its mean:

$$Z = RY - \overline{RY}$$

We then convert every vector $z \in Z$ into a signed vector representation $\hat{z}$:

$$\hat{z}_i = \text{sign}(z_i), \text{ where } \text{sign}(z_i) \begin{cases} 1 \text{ if } z_i \geq 0 \\ -1 \text{ if } z_i < 0 \end{cases}$$

Following the work from Wang et al. the probability of an anchor point $c$ and another point $y$ to be mapped into the same bit in one dimension is (Wang et al., 2015):

$$\Pr\left[h_i\left(c\right) = h_i\left(y\right)\right] = 1 - \frac{\theta_{cy}}{\pi} = 1 - \frac{1}{\pi}\cos^{-1}\frac{c^\top y}{\|c\|\,\|y\|}$$

where $h_i$ converts the vector $y$ into the binary representation $\hat{z}_i$ as described above and $\theta$ is the angle. This is illustrated in Figure 2 where we show how the cosine similarity between

the anchor and the data points affects different hyperplanes $h_i$. The Hamming distance between $c$ and $y$

$$\text{HamDist}(c,y) = \sum_{i=1}^{b} \mathbf{1}_{h_i(c) \neq h_i(y)}$$

is the number of bits they differ in. As all rows of $R$ are drawn independently it corresponds to a binomial distribution with parameters $\frac{\theta_{cy}}{\pi}$ as success rate and $b$ as number of trials. For large $b$ this is approximating a normal distribution. The smaller the angle between the data point $y$ and the anchor $c$, the more likely our method will identify $y$ as a nearest neighbor because the Hamming distance will be smaller. In our ablation study we further investigate the design choices made during the LSH process, namely the choice for an orthonormal matrix as well as the centering.

The advantages of such encoding become evident when working with large datasets. For example, reducing embeddings from $d$-dimensional 32-bit floating-point vectors, where each value requires 4 bytes, to $d$-dimensional binary representations, where each value requires only 1 bit, results in a reduction of storage requirements by a factor of 32. This drastically reduces the memory needed for storing embeddings used in HN sampling. Additionally, storing embeddings as binary vectors enables the use of Hamming distance, i.e. the number of different bits in the vectors, for similarity search, which is highly efficient due to its reliance on bitwise operations (XOR and popcount) (Wang et al., 2015). These operations are optimized in hardware, providing significantly faster similarity computations compared to cosine similarity, especially for high-dimensional data.

Table 1: Quantitative comparison between multiple sampling methods on supervised image retrieval dataset. Results are reported for Recall@1 (R@1) and Recall@5 (R@5).

| Approach | CVUSA | | $\text{CVACT}_{val}$ | | $\text{CVACT}_{test}$ | | $\text{VIGOR}_{same}$ | | $\text{VIGOR}_{cross}$ | | SOP | | InShop | |
|---|---|---|---|---|---|---|---|---|---|---|---|---|---|---|
| | R@1 | R@5 | R@1 | R@5 | R@1 | R@5 | R@1 | R@5 | R@1 | R@5 | R@1 | R@5 | R@1 | R@5 |
| Random | 97.68 | 99.63 | 87.46 | 96.46 | 60.17 | 89.35 | 64.58 | 91.19 | 36.06 | 62.96 | 87.55 | 94.70 | 91.93 | 97.92 |
| BatchHard (Schroff et al., 2015) | 97.64 | 99.63 | 87.28 | 96.65 | 60.68 | 89.46 | 66.75 | 92.28 | 36.31 | 63.71 | 87.85 | 94.93 | 91.93 | 97.93 |
| Pre-Epoch Full | 98.68 | 99.67 | 91.01 | 97.11 | 69.98 | 92.82 | 77.11 | 96.11 | 59.86 | 82.55 | 89.44 | 95.76 | 93.21 | 98.21 |
| Pre-Epoch Incr. | 98.53 | 99.62 | 90.42 | 97.12 | 68.71 | 92.50 | 76.39 | 96.01 | 57.97 | 81.61 | 89.78 | 95.75 | 93.07 | 98.30 |
| $\text{LSH}_{128}$ (ours) | 98.15 | 99.67 | 89.70 | 96.94 | 66.29 | 91.62 | 74.48 | 95.24 | 53.93 | 79.08 | 89.09 | 95.42 | 92.86 | 98.22 |
| $\text{LSH}_{256}$ (ours) | 98.43 | 99.65 | 90.07 | 97.02 | 67.27 | 91.93 | 75.50 | 95.59 | 55.99 | 80.53 | 89.34 | 95.54 | 93.19 | 98.13 |
| $\text{LSH}_{512}$ (ours) | 98.54 | 99.68 | 90.45 | 97.15 | 68.20 | 92.29 | 76.35 | 95.76 | 57.22 | 81.25 | 89.60 | 95.65 | 93.11 | 98.12 |
| $\text{LSH}_{1024}$ (ours) | 98.60 | 99.65 | 90.82 | 97.24 | 68.75 | 92.60 | 76.51 | 95.88 | 57.69 | 81.53 | 89.64 | 95.78 | 93.31 | 98.22 |

## 4. Evaluation

We compare our approach on six datasets, two from the domain of product search (Oh Song et al., 2016; Liu et al., 2016), three from the domain of cross-view geo-localization (Zhu et al., 2021; Workman et al., 2015; Liu & Li, 2019), and one textual retrieval dataset (Bajaj et al., 2016), as prototypical retrieval tasks. We compare the results on benchmark metrics, evaluating overlap and mean positional distance while assessing LSH against pre-epoch and random sampling.

### 4.1. Training Process

Our comparison uses a Siamese CNN (Chopra et al., 2005) to encode image embeddings and a Transformer (Vaswani, 2017) to generate text embeddings. To minimize computational overhead, we reuse training embeddings for pre-epoch sampling. Although these embeddings may not match the latest network updates, this approach remains efficient with minimal impact on performance. To handle the computational and memory requirements of high-dimensional embeddings, we use LSH to project them into a lower-dimensional binary space. The dimensionality of this space is determined by a random rotation matrix with bit sizes $b \in 128, 256, 512, 1024$ for our image embeddings and $b \in 128, 256, 512, 768$ for the text embeddings. After each epoch, a binary index is created and HNs are identified by computing Hamming distances. These HNs are then used in the next epoch to efficiently construct training batches.

For datasets without predefined query and reference splits, such as SOP (Oh Song et al., 2016) or InShop (Liu et al., 2016), we use a similar approach. Within each class, the hardest positive sample is selected for each positive sample based on Hamming distance.

### 4.2. Implementation Details

We use ConvNeXt-base (Liu et al., 2022) as the CNN backbone, training with a learning rate of 1E-3 and a cosine decay schedule. As Transformer we use Distill-RoBERTa-base (Sanh et al., 2019) with a learning rate of 1E-4 and a cosine decay schedule over 10 epochs. During training, we apply a weight decay of 0.01 and use label smoothing set to 0.1 to improve generalization. The InfoNCE loss (Oord et al., 2018) is used in all experiments, with a learnable temperature parameter $\tau$. All our experiments are conducted on a Nvidia DGX-2 system equipped with 16 Nvidia V100 GPUs and dual Intel Xeon Platinum 8168 processors.

### 4.3. Datasets

**CVUSA** (Workman et al., 2015), contains images from all over the US from different locations and 35,532 pairs for training and 8,884 in the validation set.

**CVACT** (Liu & Li, 2019) contains the same amount of data for training and validation, but in the region of Canberra, Australia, and extends further with a test set containing over 92k images. In CVUSA and CVACT, the street view is always centered on the aerial view.

**VIGOR** contains 90,618 aerial views and 105,214 street views with arbitrary positions within the aerial view, significantly increasing the challenge of the task (Zhu et al., 2021). These datasets provide two configurations: "cross" and "equal". In the cross setting, training data is derived from two cities, while testing is performed on the other two cities. Conversely, the same setting uses samples from all four city regions for both training and testing.

**Stanford Online Products (SOP)** (Oh Song et al., 2016) contains $\approx 120,000$ images with 22,634 different classes and nearly a 50:50 (training:testing) split.

**InShop** (Liu et al., 2016) consists of over 52k images with 7,982 different clothing types as classes.

**MS Marco** (Bajaj et al., 2016) focuses on textual retrieval, requiring the identification of relevant passages from a corpus containing 500,000 training examples and 8.8 million passages based on Bing queries.

Table 2: Quantitative comparison between multiple sampling methods on the MS MARCO dataset. Results are reported for MRR@10.

| Approach | MRR@10 |
|---|---|
| Random | 20.07 |
| BatchHard | 20.59 |
| Pre-Epoch Incr. | 26.23 |
| LSH$_{128}$ (ours) | 24.41 |
| LSH$_{256}$ (ours) | 25.67 |
| LSH$_{512}$ (ours) | 26.42 |
| LSH$_{768}$ (ours) | 26.44 |
| Pre-Epoch Full | 26.24 |

### 4.4. Sampling Strategies

For our evaluation, we compare multiple sampling methods with our proposed LSH sampling:

**Random Sampling** For the random sampling strategy random pairs are sampled, allowing HNs only by chance. The only filtering is done on the class level to prevent multiple instances of a class from being in a batch. This approach does not add computational overhead.

**Pre-Epoch Full Sampling** For pre-epoch full sampling, HNs are pre-computed before each epoch by extracting the full training dataset and selecting negatives based on cosine similarity from the similarity matrix. This sampling is the most resource intensive method, as it requires a reprocessing of the complete dataset.

**Pre-Epoch Incremental Sampling** For pre-epoch incremental sampling, HNs are extracted during training using saved embeddings before weight updates. This method is faster compared to *pre-epoch full sampling* but relies on embeddings that might partially not be updated yet.

**BatchHard Sampling** Since the loss method can influence Hard Negative Sampling (HNS) selection, we follow Schroff et al. (Schroff et al., 2015) and implement Batch-Hard for the InfoNCE loss. BatchHard calculates the loss using only the 50% hardest negatives within a batch.

### 4.5. Impact of Different Sampling Strategies

We evaluate all sampling strategies on the used datasets. As shown in Section 3.2, random sampling consistently underperforms any form of sampling. While BatchHard sampling can achieve higher performance, retaining only 50% of the HNs results in marginal improvements, since HNs are not explicitly selected. In addition, BatchHard Sampling has the disadvantage of discarding some computed results because

it artificially limits the number of HNs used.

Pre-Epoch Full Sampling, while the slowest approach, often yields the best performance since embeddings are extracted after each completed epoch. This allows the model to generalize effectively throughout the training process.

Comparing Pre-Epoch Incremental and LSH, we obtain slightly worse performance when the bit dimension is low (128 or 256) and the same performance when the bit dimension is high (512 or 1024), while being faster and requiring less space to store the vector embeddings.

Similar results are observed for text retrieval, see Section 4.3, where random sampling underperforms, and HN improves performance. Unlike vision tasks, higher LSH bit counts further boost performance over Pre-Epoch full sampling.

Additionally, we compare the speed and space costs of the pre-epoch incremental and LSH sampling for searching and calculation. Furthermore, we investigate the relationships between sampled neighbors by LSH and the actual hardest negatives determined via the cosine similarity matrix.

### 4.6. Search Speed Comparison

In Figure 1, we present a comparison of search times between our LSH-based feature encoding with different bit sizes (128, 256, 512, 768, 1024) and HN sampling using float32 vector embeddings. To ensure a fair evaluation, we used the FAISS library (Douze et al., 2024) to retrieve the top 128 Nearest Neighbor (NN) in each configuration.

Although the theoretical query time for all indices is $O(n \cdot d)$, the use of binary features (LSH) leads to significant speedups. For this experiment, we encoded the training data of each dataset, except CVACT, since its size is identical to that of CVUSA. For datasets with a reference-query split, we performed the search within the query set to retrieve 128 NN for each reference.

The gap between LSH-based coding and full vector embeddings remains consistent across datasets, including large datasets such as MS MARCO with over 500,000 samples, where the search time remains significantly shorter compared to indices that compute cosine similarity with full precision. The reported times represent the duration required per epoch during training for searching, resulting in a significant impact on the overall training time.

### 4.7. Neighbor Analysis

We further investigate the behavior of HNs selection when LSH is used. For this analysis, we use the MS-MARCO dataset with over 500k samples and the SOP dataset, which contains over 60k samples, to comprehensively evaluate the generalization of our approach. In our appendix results for the VIGOR dataset can be found as well. In each setting,

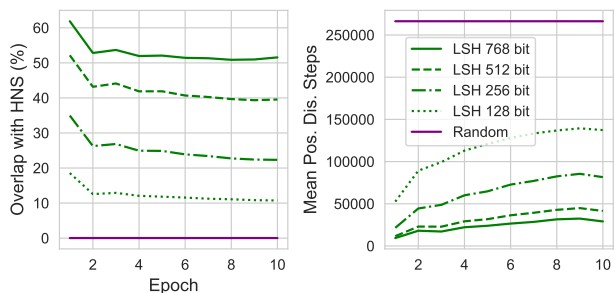

Figure 3: A comparison of overlap and mean positional distance of LSH with varying bit sizes (128, 256, 512, and 1024) and random sampling. The overlap with Pre-Epoch Increment (HNs) and mean positional distance (right) on MS MARCO.

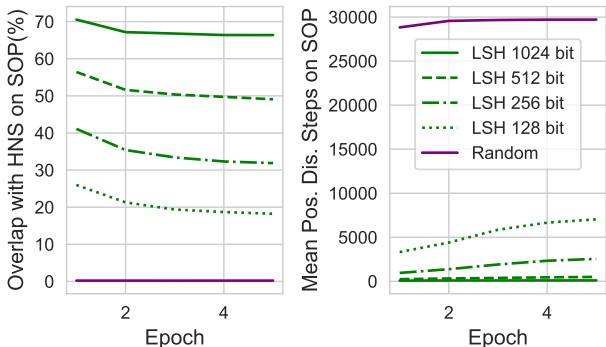

Figure 4: A comparison of LSH with varying bit sizes (128, 256, 512, and 1024) and random sampling is presented across two metrics: overlap with Pre-Epoch Increment (HNs) and mean positional distance (right) on SOP.

we retrieve the top 128 NNs and rank them based on their similarity to the reference feature. This allows us to compute the mean positional distance between the ANNs obtained by LSH and the actual NNs determined by cosine similarity. Furthermore, we evaluate the overlap between the retrieved neighbors and the real nearest neighbors to quantify the effectiveness of the approximation.

### 4.7.1. NEIGHBOR OVERLAP

We compare the overlap between the examples used in a batch and those retrieved based on cosine similarity. For SOP, as shown Figure 4, the overlap with random sampling is approximately one percent. In contrast, using LSH significantly increases the overlap, and our strategy achieves around 70% overlap at the highest bit count.

Based on the evaluation presented in Section 3.2 and the overlap for 54% for 512 bits, this precision appears sufficient to provide enough HNs for achieving satisfying performance. It is interesting to note that especially in lower bit regimes (128 and 256) the overlap reduces over time. This may seem counterintuitive at first, but it is consistent with the goal of the loss function. The loss function is designed to encourage negative samples that are close neighbors to become more dissimilar. As a result, it becomes increasingly difficult to identify real HNs, especially considering that the probability of selection depends on similarity, as described in Section 3.2. In the mean positional distance plot (see Figure 4), random sampling converges to the center of the dataset. This is expected, as positional relationships do not influence its selection. In contrast, the 1024-bit LSH remains very close to zero, with fluctuations increasing as the number of bits decreases.

When analyzing the textual modality, we can observe a different behavior, as shown in Figure 3. The overlap is notably lower compared to the vision task, and the mean positional

distance remains far from zero, even with higher bit counts. While the real HNs can still be identified, the process for the textual modality is considerably more challenging compared to the vision task. This may stem from the nature of text, which captures multiple concepts and fine-grained distinctions, unlike the broader and more cohesive concepts typical of images. Nevertheless, 512 bits still provide strong performance, even surpassing pre-epoch incremental and full sampling.

Furthermore, we compare the cosine similarity between the retrieved and actual HNs, with details in the appendix.

### 4.7.2. NEIGHBOR HARDNESS

The question of how many HNs (HNs) are required to maintain robust performance in HN sampling is crucial to understanding the trade-offs associated with using ANN. As detailed in Section 4.7, the LSH algorithm achieves approximately 70% overlap with real HNs. This study evaluates how performance improves when more hard samples are used and hardness defines the percentage in a batch that is a HN. Figure 5 shows the results of training for 10 epochs with different levels of hardness. A hardness level of 1.0 represents the setting used for pre-epoch incremental sampling where always all HNs are retrieved and 0.0 represents random sampling. We also include our LSH results based on the overlap from Figure 4. While the performance on SOP remains similar, our method significantly improves results on MS Marco, even at the same hardness levels. This demonstrates that LSH selects more effective HN, even if they are not the real true HN, compared to random selection. For VIGOR, we present the same experiment in the appendix.

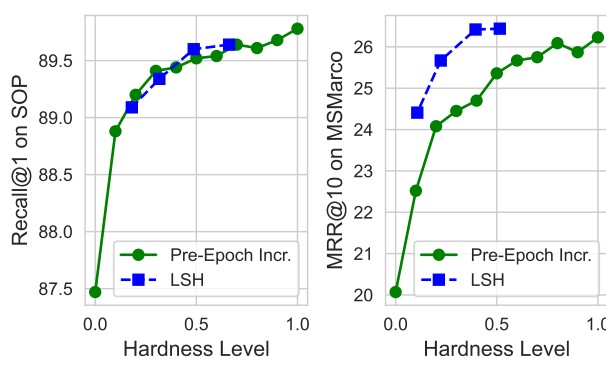

Figure 5: Impact of HN hardness on R@1 on SOP (left) and MRR@10 on MSMarco (right). We define hardness as the percentage of HNs within a batch and include the results from LSH based on the respective overlap.

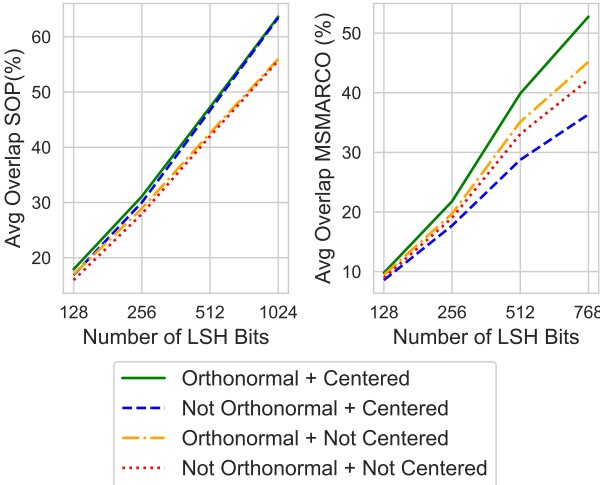

Figure 6: Impact of our LSH design choices on the overlap for SOP and MS MARCO.

### 4.8. LSH Design Choices

In Section 4.7.1 we showed how a smaller bit size results in less overlap with the actual NN, and as we can see in our performance evaluation, Section 3.2, less overlap results in worse benchmark scores on the dataset. We now want to investigate how the design choices of our method improve this overlap. As described in Section 3.2, we use a random rotation matrix with orthonormal vectors and center the projected features. To obtain orthonormal vectors, we use QR decomposition. As shown in Figure 6, centering consistently improves the overlap between the found NN and the actual NN, especially in lower dimensional hash spaces, by reducing the skew in the binary representations.

For the MS MARCO dataset, the use of orthonormal matrices becomes increasingly important at higher bit dimensions, preserving feature variance and improving alignment with neighbours retrieved with the cosine similarity. Without orthonormalization, overlap performance degrades, especially in high-dimensional spaces.

### 5. Conclusion

We show that LSH, through its inherent properties such as locality preservation and similarity-based retrieval, effectively approximates real NNs and enables robust HNs mining. Our experiments show that even at lower bit sizes (e.g., 256 or 512), the sampled neighbors achieve strong overlap with real NNs and generalize better than traditional methods such as BatchHard (Schroff et al., 2015) or random sampling. This confirms that LSH can serve as reliable and efficient technique for HN sampling in contrastive learning.

Further, we show that the use of LSH significantly reduces the time and space complexity associated with traditional pre-epoch sampling. The binarization process naturally re-

duces memory requirements, while the slower scaling of search time allows our approach to handle larger datasets more efficiently compared to exact search methods. Despite these reductions, the performance and quality of HN selection remain competitive or performs even better, further validating the practical advantages of our method.

### 6. Discussion

In our work, we focus on supervised learning and acknowledge a key challenge in finding HNs: positive examples and identified HNs may belong to the same underlying class, particularly in unsupervised settings. This overlap complicates model convergence. While this issue has been explored in related literature (Robinson et al., 2020; Chuang et al., 2020), we do not address it further here, leaving it as a limitation for future research.

As explained in Section 4.7.2 the overall quality of our selected ANN delivers solid experimental results. Nonetheless, providing a quality guarantee for the identified HNs would be advantageous, similar to the ANN guarantees in (Har-Peled et al., 2012). Furthermore, it remains an open question whether theoretical bounds on the performance of the final embedding can be derived from such quality guarantees for the ANN.

Another interesting question remains the comparison of our LSH-based approach with other ANN methods like Product Quantization (Jégou et al., 2011) or Hierarchical Navigable Small World graphs (Fu et al., 2019).

## Impact Statement

This work proposes a more efficient sampling approach for contrastive learning that focuses on approximating HNs to enable faster training and reducing computational cost. We demonstrate the effectiveness of the method on general retrieval tasks such as dense passage retrieval and product search. Recognizing the ethical concerns of person re-identification, we deliberately avoid using such datasets and limit our engagement with this topic to a literature review. This work aims to advance contrastive learning while promoting responsible and sustainable research practices.

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

# A. Appendix

## A.1. Similarity Distribution Analysis

We further examine the distribution of similarities between the ANN identified by our sampling method and the real NN retrieved by cosine similarity, as shown in Figure 7. Notably, while random sampling produces a uniform similarity distribution, increasing the bit count consistently shifts the distribution toward 0.5, regardless of the dataset used. This highlights the advantage of using LSH, as the similarity in the original embedding space affects the probability of hash collisions. Even if the true NN is not found, the retrieved examples are better than those obtained by random sampling.

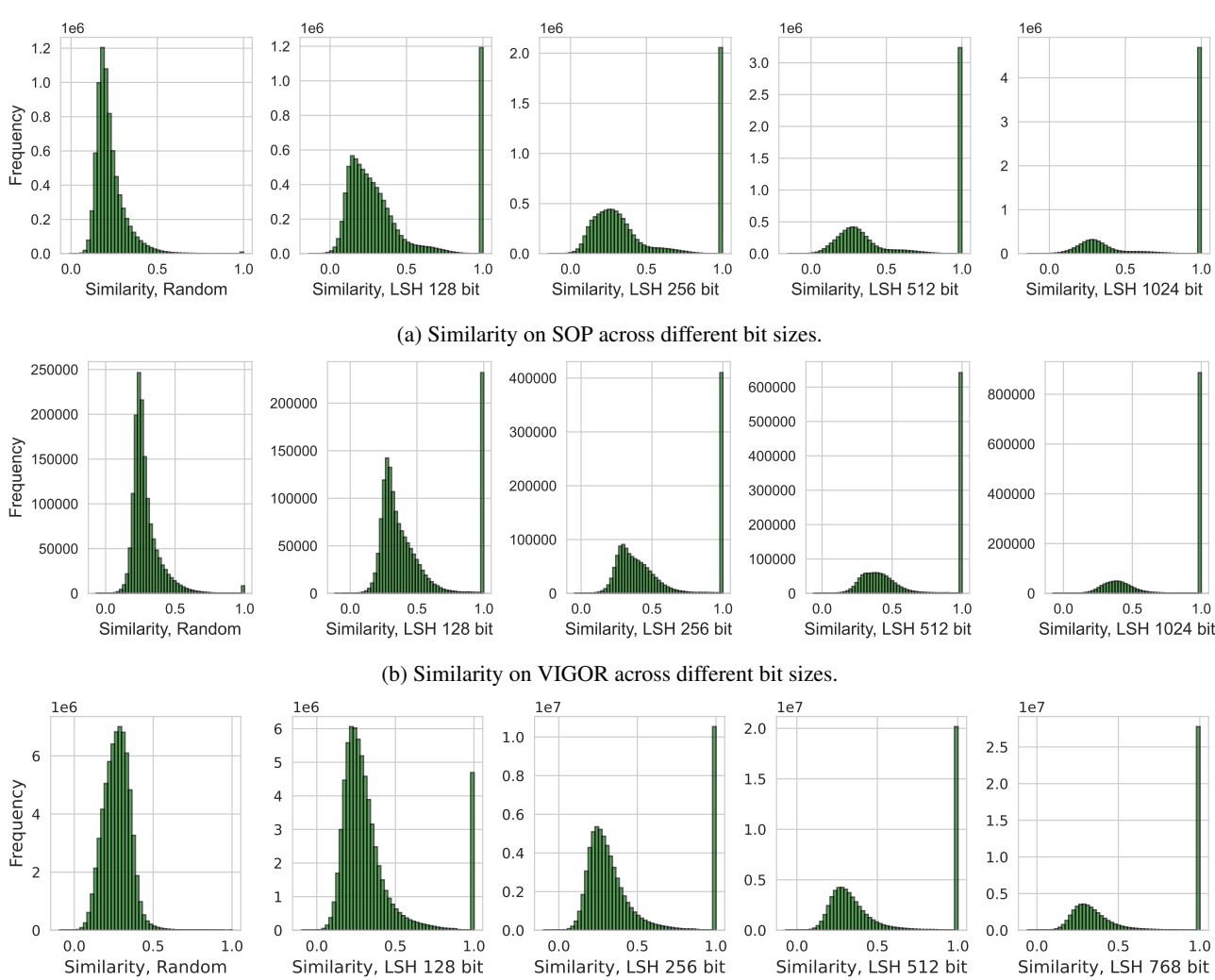

(a) Similarity on SOP across different bit sizes.

(b) Similarity on VIGOR across different bit sizes.

(c) Similarity on MS MARCO across different bit sizes.

Figure 7: Comparison of the similarity between the retrieved approximated HNs and the actual HN retrieved by the cosine similarity for SOP (top), VIGOR(middle) and MS MARCO (bottom).

## A.2. VIGOR Analysis

Furthermore, we also compare the overlap and mean positional distance a subset of VIGOR in Figure 8. In this subset we only use the city of Seattle for training and evaluation. Similar to the other datasets the overlap declines over time as the embeddings of pairs are pushed afar from each other. We also investigate the impact of hardness during training in Figure 9, increasing hardness of the sample improves the Recall@1 on the VIGOR dataset. We further include the results achieved with LSH based on our overlap depicted in Figure 8. Similar to MS MARCO we achieve higher values of Recall@1 while

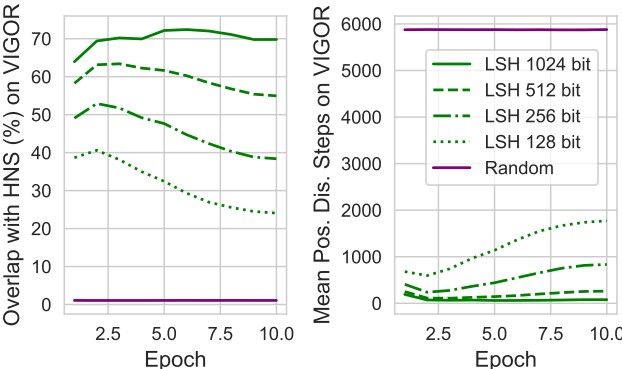

Figure 8: A comparison of overlap and mean positional distance of LSH with varying bit sizes (128, 256, 512, and 1024) and random sampling. The overlap with Pre-Epoch Increment (HNs) and mean positional distance (right) on VIGOR.

the overlap remains the same.

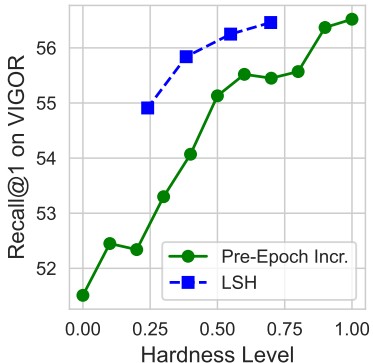

Figure 9: Impact of HN hardness on R@1 on VIGOR. We define hardness as the percentage of HNs within a batch and include the results from LSH based on the respective overlap.

### A.3. Further Implementation Details

For our image-based experiments, we apply several data augmentation techniques during training, including flipping, rotation, coarse dropout, grid dropout, and color jitter. These augmentations help improve model generalization by introducing variability into the training samples.

In supervised settings, where multiple positive pairs exist for a given label, we structure our batches to contain only one positive pair per label. This approach minimizes redundancy and reduces noise in the loss computation, ensuring a more stable training process.

We train for 40 epochs on the CVUSA, CVACT, and VIGOR datasets, which focus on cross-view geo-localization tasks. For datasets with a different retrieval structure, such as Stanford Online Products (SOP), InShop, and the MS MARCO text retrieval dataset, we limit training to 10 epochs to avoid overfitting. For the cross-view dataset, we resize the images for CVUSA and CVACT to $384 \times 384$ for the satellite image and $112 \times 616$ for the street view image, for VIGOR we use $384 \times 384$ for the satellite view and $384 \times 768$ for the street view. In the experiments for SOP and InShop, all images are resized to $384 \times 384$.

### A.4. Training Process:

Figure 10 illustrates the process of encoding arbitrary input data, such as text or images, using an encoder to generate embeddings. LSH is applied to transform these embeddings into binary vectors. After each epoch, pairwise search based on Hamming distance is used to sample HNs, for the next epoch. Similar to sampling on the float32 embedding, we define the HN as the negative sample with the smallest distance to the anchor.

### A.5. Architecture Details:

For our experiments on image datasets, we use the ConvNeXt base model (Liu et al., 2022), pre-trained on ImageNet-21k, from the timm library (Wightman, 2019). ConvNeXt modernizes the ResNet architecture by incorporating design principles from the Vision Transformer (Dosovitskiy, 2020). The model outputs 1024-dimensional embeddings and consists of 88 million parameters.

For our experiments on the MS MARCO text retrieval dataset, we use Distill-RoBERTa-base (Sanh et al., 2019), a distilled version of RoBERTa (Liu, 2019), with 82 million parameters. The hidden size of the transformer is 768. In both cases, we choose these models to allow efficient training and evaluation of our methodology. Additionally, we employ shared weights for both reference and query inputs.

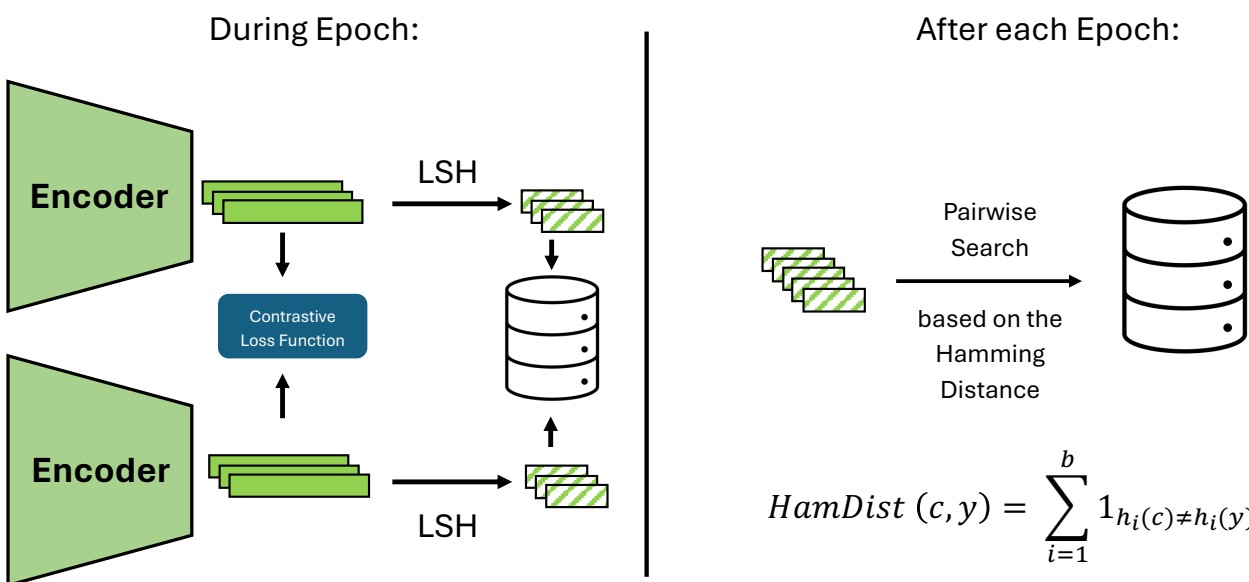

$$HamDist\,(c, y) = \sum_{i=1}^{b} 1_{h_i(c) \neq h_i(y)}$$

Figure 10: Framework for encoding input data and leveraging LSH for binary transformation and hard negative sampling during contrastive learning.