# OpenReview forum: "Locality-Sensitive Hashing for Efficient Hard Negative Sampling in Contrastive Learning"
_ICML.cc/2025/Conference — Submitted to ICML 2025_

### Official Review · Reviewer_s7Gy · 2025-03-14

**Overall Recommendation:** 3

**Summary:**

This paper addresses the computational challenge of efficiently finding high-quality hard negative (HN) examples in large, high-dimensional datasets for contrastive learning. The authors propose a novel GPU-friendly Locality-Sensitive Hashing (LSH) technique which projects the input vectors to binary buckets, then apply xor + popcount to find neighbors with the smallest hamming distances. The approach is evaluated on several textual and visual datasets, demonstrating superior computational efficiency compared to other HN mining strategies without significant performance degradation. The key contribution is a lightweight framework for HN sampling that leverages a global view of the dataset while maintaining low computational costs during training.

**Claims And Evidence:**

Below are the two major claims from the paper:
1. Built a lightweight and efficient framework for HN sampling using LSH, offering a global view of the dataset with low computational costs. Authors provided end-to-end runtime in Figure 1. However, a LSH vs kNN or other approximated nearest neighbor search benchmark is missing.
2. Demonstrated mining pre-epoch hard negatives using the LSH method above improves the model performance significantly. Authors evaluated the idea on six datasets across two modalities.

**Essential References Not Discussed:**

The xor-popcount computation can be written in regular CUDA programs, but on the Nvidia Ampere architecture there's a specialized tensor core instruction (XorPopc) that can compute much faster than regular CUDA programs (https://github.com/NVIDIA/cutlass/blob/main/include/cutlass/arch/mma_sm80.h#L1441C3-L1441C12).

**Experimental Designs Or Analyses:**

This work compared against random sampling, batch hard sampling, pre-epoch full sampling, and pre-epoch incremental sampling. Demonstrating competitive results to full sampling, but with better runtime. Authors also alter different numbers of bits and observe the mean positional distance of the HSH estimate.

However, it would be great if the authors can also provide a recall-qps trade-off plot like many other ANN research does. https://ann-benchmarks.com/index.html.

The other thing that is missing is the original embedding size used for nearest neighbor search. The most accurate LSH setting provided in this paper used 1024 bits, which may not be that small when compared to the original embedding size.

**Methods And Evaluation Criteria:**

This paper introduces a method employing Locality-Sensitive Hashing (LSH) as an Approximate Nearest Neighbor (ANN) technique to mitigate the computational overhead associated with pre-epoch Hard Negative (HN) sampling. The proposed LSH method projects input vectors using random orthogonal bases and subsequently quantizes them into binary representations, encoding the position of each datapoint relative to hyperplanes. This binarization process substantially decreases storage requirements and facilitates rapid similarity searches via Hamming distance, which leverages efficient bitwise operations (XOR and popcount). During training, anchors are iteratively sampled, along with their corresponding positive and negative sample batches. Positive samples are drawn closer to the anchor, while negative samples are pushed away, thereby promoting accelerated convergence. Among the samples, those exhibiting the closest distance yet classified as negative to the anchor are deemed "hard negatives" and are critical for training efficacy. The authors propose the utilization of their LSH technique to expedite the identification of these hard negatives, employing a GPU-optimized LSH implementation that utilizes binary operations of XOR and popcount.

The methodology was evaluated across six datasets and benchmarked against random sampling, batch hard sampling, pre-epoch full sampling, and pre-epoch incremental sampling. Results demonstrate a significant reduction in runtime compared to the brute-force pre-epoch approach, while achieving a level of quality comparable to pre-epoch full sampling.

**Other Comments Or Suggestions:**

-

**Other Strengths And Weaknesses:**

-

**Questions For Authors:**

1. Can author provide the recall-qps trade-off chart like https://ann-benchmarks.com/index.html?
2. Can author provide some theoretical analysis of how the hamming distance relates to the ground truth nearest neighbors?
3. What is the size of the original embedding? Can authors plot the compression ratio of LSH vs its recall in a chart as well?

**Relation To Broader Scientific Literature:**

-

**Theoretical Claims:**

The theoretical claims of hard negative mining follows previous work in hard negative minings.

The random projection followed by hyperplane splitting also follows previous work. However, it may be better to provide some proofs and insight of how the projected hamming distance relate to the original k nearest neighbors.

---

> ### Author Rebuttal · Authors · 2025-03-31
>
> 1. Thank you for your suggestion. We agree that including a recall-QPS trade-off chart for our LSH approaches would provide additional clarity. We will replace Figure 1 with this graph. While Figure 1 was originally intended to illustrate processing time from a dataset size perspective by comparing our LSH method to cosine similarity over full embedding sizes, we believe the proposed chart provides a clearer and more direct view of the speed-performance tradeoff of our approach.
> 2. Yes, we have given a more detailed insight to this topic in answer 3 for reviewer udPs.
> 3. The original embedding depends on the data modality. For our image retrieval task we utilized a ConvNeXt-Base thus resulting in an embedding size of 1024 of float32 values. For the textual modality we utilized a Distill-RoBERTa-base with a embedding size of 768 of float32 values, we add this in 4.2. Implementation Details. While we describe the compression ratio briefly in Line 205-210 we agree with the reviewer that this should be better visualized to further strengthen our proposition, we can add a plot for this in our supplementary.
>
> We also thank the reviewer for making us aware that cuda supports  xor-popcount as a native implementation. This enables an even faster calculation of our methodology for the search directly on the GPU, which further accelerates our methodology!

---

### Official Review · Reviewer_TJqN · 2025-03-14

**Overall Recommendation:** 2

**Summary:**

This paper explores hard negative (HN) sampling in contrastive learning and proposes a Locality-Sensitive Hashing (LSH)-based Approximate Nearest Neighbor (ANN) approach to improve computational efficiency while maintaining competitive performance. The proposed method enables fast and efficient pre-epoch HN selection, making it scalable to large datasets. The paper includes experiments on multiple textual and visual datasets to evaluate its effectiveness. While the research topic is relevant and well-motivated, several methodological and presentation issues need to be addressed to strengthen the validity and clarity of the work.

## update after rebuttal

**Claims And Evidence:**

Yes

**Essential References Not Discussed:**

no

**Experimental Designs Or Analyses:**

Yes

**Methods And Evaluation Criteria:**

Yes

**Other Comments Or Suggestions:**

No

**Other Strengths And Weaknesses:**

Strengths:
1. The paper addresses an important challenge in contrastive learning—efficient hard negative (HN) sampling. By leveraging Locality-Sensitive Hashing (LSH) as an Approximate Nearest Neighbor (ANN) approach, the study provides a scalable solution that is particularly relevant for large-scale datasets.
2. The proposed method significantly reduces the computational cost of HN selection, making it more practical for real-world applications where dataset size is a bottleneck.
3. The paper evaluates the method on **both textual and visual datasets, which strengthens its applicability across different modalities.

Weaknesses:
1. The paper lacks a strong mathematical foundation for key claims, such as the correlation between cosine similarity and Hamming distance. Providing a formal derivation or empirical validation would enhance the rigor of the study.
2. Some parts of the paper, particularly mathematical formulations and explanations of feature transformations, are unclear and require better notation, proper equation numbering, and detailed symbol definitions.
3. While LSH is known for its efficiency, the paper does not adequately compare it with alternative dimensionality reduction techniques (e.g., PCA, learned embeddings) or discuss how much information is lost in the process.

**Questions For Authors:**

1. The current structure of the paper contains redundancies and sections that could be streamlined to enhance readability.
1) Such as "we propose a lightweight Approximated Nearest Neighbor (ANN) approach that leverages Locality-Sensitive Hashing (LSH)" and "Our work first explores HN sampling methods and introduces LSH as an ANN approach" are repetitive.
2) Preliminary section (3.1) contains excessive common knowledge. Large portions of this section summarize well-known aspects of contrastive learning and hard negative mining, making it longer than necessary.

2. Justification of Hash Encoding Needs Clarification. The paper proposes LSH-based binary encoding to reduce computational cost, but it does not clearly explain its advantages over other dimensionality reduction techniques.
1) What makes LSH particularly suitable for this problem compared to traditional PCA-based or learned feature compression methods?
2) Does the hashing step introduce any loss of information? If so, how does it affect the quality of hard negative mining?

3. The Purpose and Impact of Data Transformations (Lines 197-208) Need More Explanation. The paper describes a series of transformations applied to the original feature representations but does not provide a clear justification for them. Why are these specific transformations applied? Is there empirical evidence that these steps improve the quality of HN sampling?

4. Several equations in the paper lack clarity, proper notation, and explanations of symbols, making them difficult to follow.
1) Equations should be numbered and properly punctuated. Example: Lines 206 and 214 introduce equations without explaining all variables used (e.g., what does "i" represent?).
2) The paper discusses random rotation matrices (R) and binary hashing functions (hi) but does not clearly define their mathematical properties. Are R and hi learnable or fixed?

5. The paper claims that angle-based similarity (cosine similarity) and binary Hamming distance are positively correlated, but it does not provide a theoretical proof or strong empirical validation for this assumption. What is the precise mathematical relationship between cosine similarity and Hamming distance?

6. Is the Feature Mapping Learnable? The transformation of original feature vectors into binary hash codes raises concerns about potential information loss. Are the transformation matrices (R) and hashing functions (hi) optimized during training, or are they static? If they are static, how do we ensure that important information is retained?

**Relation To Broader Scientific Literature:**

Yes

**Theoretical Claims:**

Yes

---

> ### Author Rebuttal · Authors · 2025-03-31
>
> We thank the reviewer for the comments and will address them accordingly:
> 1. Thank you for the suggestion, we will address this and structure the paper more clearly by, for example, removing redundancies and improve citation clarity as suggested by udPs.
> 2. See 1.
> 3. See 1.
> 4. LSH has the advantage of not being data dependent. We think that data dependent feature compression methods such as PCA would fail in this task. First, they often assume some underlying data structure. PCA for example needs the data to be distributed in some linear subspace. As the whole purpose is to find a suited embedding space it seems counterintuitive to us to add arbitrary constraints to this embedding. Second, data dependent compression methods by definition depend on the current embedding. That means as the embedding space changes over time, the compression model must be refitted. Recalculating the SVD for PCA after every update is computationally expensive. LSH on the other hand is a very lightweight and efficient method for the sole purpose of finding an ANN.
> 5. See 4.
> 6. Yes, the hashing step introduces a loss of information due to both dimensionality reduction and binarization of the feature vectors. We empirically analyze how this affects the quality of hard negative sampling in Figures 3 and 4. In particular, we observe that as the number of bits decreases, the overlap with cosine similarity-based hard negative sampling decreases and the average positional distance to retrieved negatives increases. Furthermore, our quantitative results (Tables 1 and 2) confirm that lower bit counts correspond to performance degradation when using LSH-derived negatives compared to cosine similarity-based hard negative sampling.
> 7. Thank you for pointing out that there is still some clarity missing. We follow the classical procedure of LSH with random projections for ANN as in Wang et al. 2015 (Learning to Hash for Indexing Big Data - A Survey) which we will better explain in Section 3.2.
> 8. We will improve the structure and clarity in the math notations in the corresponding Section 3.
> 9. Thank you for pointing this out. We will double check our equations and add additional explanations as suggested. In this particular case “i” refers to the index of vector z. We wanted to describe that every dimension of the projection z of and embedding y is converted into a bit depending on its sign.
> 10. We state that we use a random rotation matrix that features orthonormal vectors sampled from a gaussian distribution and is fixed throughout the training. The gaussian sampling is taken from Wang et al. 2015 (Learning to Hash for Indexing Big Data - A Survey)  and orthonormalization is empirically justified by our experiments. Thus, the matrix R is not learnable. $h_i$ should denote the binarization of the affine linear transformation of the ith row of the random rotation matrix and is thus not learnable too.
> 11. According to Wang et al. 2015 (Learning to Hash for Indexing Big Data - A Survey)  the collision probability of two vectors, c and y in the ith bucket $(i.e. P(h_i(c)=h_i(y)))$ is equal to $1 - \frac{\theta_{cy}}{\pi}$, $\theta_{cy}$ denoting the angle between c and y. As the hamming distance is just the sum of dependent Bernoulli trials, each of them with a success rate of $1-P(h_i(c)=h_i(y))=1 - \frac{\theta_{cy}}{\pi}$ it equals to the said Binomial distribution. We explained this in more detail in answer 3 for reviewer udPs.
> 12. We only learn the embedding. The procedure to identify hard negatives is static throughout the whole training process as mentioned in Section 4. It is solely based on the random initialization of the projection matrix. Information loss is not that relevant as the only necessary condition is that points with high cosine similarity result in a low hamming distance between them. Preserving additional structure or information is not necessary, as our experiments show, and we will add a statement of that effect in 3.2 explaining our approach.

---

### Official Review · Reviewer_XybM · 2025-03-14

**Overall Recommendation:** 2

**Summary:**

This paper introduces a Locality-Sensitive Hashing-based method for efficient Hard Negative sampling in contrastive learning. This method converts feature embeddings into binary representations, which enables fast approximate nearest neighbor searches.

**Claims And Evidence:**

Most of the claims are supported by experiment results. However, this paper lacks formal theoretical guarantees on the quality of LSH-based hard negation and does not analyze the impact of different hash function choices.

**Essential References Not Discussed:**

N/A

**Experimental Designs Or Analyses:**

I suggest that the authors could add some experiments to analyze the effect of dataset size on LSH performance.

**Methods And Evaluation Criteria:**

The proposed method and evaluation criteria make sense.

**Other Comments Or Suggestions:**

Please refer to above all parts.

**Other Strengths And Weaknesses:**

Strengths:
1. This paper is well-structured and well-written.
2. The experiments are comprehensive and include various datasets.

Weaknesses:
1. The authors could consider providing formal error bounds for LSH-based sampling. It would be helpful to analyze how approximation errors affect model performance. Comparing LSH errors with other approximate nearest neighbor methods could also strengthen the evaluation.
2. There is no formal theoretical analysis for LSH-based hard negative sampling. I think adding a mathematical guarantee or error analysis would make the paper more rigorous.

**Questions For Authors:**

1. I'm wondering if LSH could adjust bit sizes dynamically during training based on model convergence to optimize both efficiency and accuracy. Have you explored any adaptive strategies for selecting the optimal bit size at different training stages?
2. Would LSH performance degrade in highly structured feature spaces where feature relationships follow a specific hierarchy? If so, what measures can be taken to mitigate potential issues and ensure robust performance across different data distributions?

**Relation To Broader Scientific Literature:**

This work is based on contrastive learning and approximate nearest neighbor search literature.

**Theoretical Claims:**

In this paper, the authors didn't introduce formal proofs for theoretical claims. Specifically, they didn't provide formal bounds for the probability that an LSH-sampled neighbor is a true hard negative and they didn't explore the impact of different hash functions on retrieval quality.

---

> ### Author Rebuttal · Authors · 2025-03-31
>
> We thank the reviewer for the comments and will address them accordingly:
> 1. Yes we agree with Reviewer XybM, that investigating dynamic bit sizes can be interesting. Therefore we already explored this idea on both the VIGOR dataset and MS MARCO. We started with 8 bits and gradually increased to 16, 32, 64, up to 1024 bits over the training epochs. However, this adaptive strategy resulted in slightly worse performance than using a fixed 128-bit setting throughout training. This effect appears to be due to the interaction with the cosine decay learning rate schedule. As training progresses, higher bit resolutions allow for finer-grained hard negative mining, which increases the difficulty of the retrieved samples. At the same time, the decreasing learning rate limits the model's ability to adapt to these harder samples, resulting in smaller weight updates. In principle, this problem could be mitigated by combining bit increase with a dynamic learning rate schedule (like multi-cycle or restart-based), but how to determine the increase of the needed learning rate is an open question. We agree that this is a promising direction for future work and can include the experiments we already did in the supplementary material.
> 2. Performance degradation of LSH in highly structured feature spaces is unlikely. In fact, structured data should typically exhibit high cosine similarity between embeddings belonging to the same hierarchical cluster. Given $$ \operatorname{Pr}\left[h_i\left(c\right)=h_i\left(y\right)\right]=1-\frac{\theta_{c y}}{\pi}=1-\frac{1}{\pi} \cos ^{-1} \frac{c^{\top} y}{\left\|c\right\|\left\|y\right\|} $$ , data points that are hierarchically or structurally related will naturally have a high probability of sharing identical or closely matching hash vectors. This property ensures the robustness of LSH when dealing with structured or hierarchical data distributions. Moreover, the slight randomness inherent in LSH can actually enhance contrastive learning by occasionally sampling negatives from nearby but different clusters, thus improving batch diversity and generalization.
>
>
> We have provided an empirical insight into the relationship between the quality of LSH samples and model performance in figure 5. From a theoretical viewpoint Har-Peled et al. 2012 (Approximate Nearest Neighbor: Towards Removing the Curse of Dimensionality) work on a slightly different problem and Wang et al. 2015 (Learning to Hash for Indexing Big Data - A Survey) which we base our LSH paradigm on does not provide any deeper theoretical insides. Therefore we could not fall back on their work. The conclusion we draw regarding the implications on the hamming distance will be reworked to provide a clearer and better structural insight. We explain this in more detail in answer 3 for reviewer udPs.

---

### Official Review · Reviewer_udPs · 2025-03-17

**Overall Recommendation:** 1

**Summary:**

The paper addresses the efficient sampling of hard negatives in contrastive learning by introducing an approximate nearest neighbor method based on Locality Sensitive Hashing (LSH). This method quantizes real-valued feature vectors into binary representations for approximate nearest neighbor search, thereby reducing both search time and space costs. The method is demonstrated to outperform other hard negative mining strategies in terms of computational efficiency on multiple text and visual datasets, with no significant degradation in performance.

**Claims And Evidence:**

1. The paper conducts experiments across multiple datasets, comparing the search times of the LSH method with other methods (such as random sampling, BatchHard sampling, Pre-Epoch Full sampling, and Pre-Epoch Incremental sampling). The results show that the LSH method significantly reduces search time under different bit-width settings.
2. The paper evaluates performance on multiple datasets using metrics such as Recall@1, Recall@5, and MRR@10. The results indicate that the LSH method maintains computational efficiency while achieving performance comparable to or better than the best methods.
3. The paper performs experiments on multiple text and visual datasets, including MS MARCO, CVUSA, CVACT, VIGOR, SOP, and InShop. The results demonstrate that the LSH method performs well on both modalities.

**Essential References Not Discussed:**

1. The paper proposes an efficient hard negative sampling method based on LSH, but does not cite some early works that applied LSH in contrastive learning.
2. The paper conducts a theoretical analysis of the LSH method, but does not reference some important literature on the theoretical foundations of LSH.

**Experimental Designs Or Analyses:**

1. Figures 4 and 5 demonstrate that LSH can effectively approximate true nearest neighbors by overlapping rate and average position distance. Figure 7 further shows that the similarity distribution of LSH retrieved samples outperforms random sampling.
2. Tables 1 and 2 present the Recall and MRR@10 metrics of the LSH method on multiple datasets, which are close to or better than the Pre-Epoch Full. However, the storage savings are only mentioned through theoretical analysis, and no actual memory usage experimental data is provided.
3. The paper conducts experiments on both image (e.g., CVUSA, SOP) and text datasets (e.g., MS Marco), but the performance on the text modality is relatively weaker. The paper does not deeply analyze how the inherent differences between text and image features affect the performance of LSH.

**Methods And Evaluation Criteria:**

The paper proposes an efficient sampling method based on LSH, which significantly reduces computational and storage overhead by quantizing high-dimensional feature vectors into binary representations. The paper uses standard retrieval metrics such as Recall@1, Recall@5, and MRR@10 to evaluate the model's performance. These metrics are widely used in information retrieval and contrastive learning tasks and effectively measure the model's performance in retrieval tasks. Therefore, the method and evaluation criteria proposed in the paper are suitable for the current problem and application, especially in large-scale datasets and high-dimensional feature spaces, where the LSH method demonstrates significant advantages.

**Other Comments Or Suggestions:**

1. It is recommended to supplement more literature in the related work section, especially the important progress in the fields of contrastive learning and LSH in recent years. This will help readers better understand the background and contributions of the paper.
2. It is recommended to unify the citation format of references to enhance the professionalism and readability of the paper.
3. It is suggested to further deepen the theoretical analysis of the LSH method, which will help strengthen the theoretical foundation of the paper.

**Other Strengths And Weaknesses:**

1. The paper proposes an efficient hard negative sampling method based on LSH, which is a significant addition to existing contrastive learning methods. This approach significantly reduces computational complexity by quantizing high-dimensional feature vectors into binary representations while maintaining the quality of hard negative samples.
2. Hard negative samples play a crucial role in contrastive learning, significantly enhancing model performance. The method not only improves sampling efficiency but also reduces computational resource consumption, which is significant for processing large-scale datasets.
3. LSH has been extensively studied in approximate nearest neighbor search, and the method in this paper primarily applies it to hard negative sampling in contrastive learning, lacking deeper innovation.
4. The paper has inconsistent citation formats and the structure is not clear enough.

**Questions For Authors:**

1.The paper mentioned applying LSH to hard negative sampling in contrastive learning. However, LSH has already been extensively studied for approximate nearest neighbor search. Could you further elaborate on the innovative aspects of your method compared to existing work?
2.The paper notes that nearest neighbor retrieval for textual data is more challenging than for image data, likely due to the semantic complexity and polysemy of text. Have you considered designing specialized LSH strategies for different modalities?
3.Despite the efficiency of LSH in approximating nearest neighbors, the paper mentions the lack of theoretical guarantees for the quality of hard negative sampling. Have you considered providing a more rigorous theoretical analysis for the LSH sampling method?
4. Why not using other more recent data-independent hashing methods to help the hard negative sampling problem?

**Relation To Broader Scientific Literature:**

1. The paper utilizes Locality-Sensitive Hashing (LSH) for efficient hard negative sampling, building on the widespread application of LSH in reducing computational complexity in large-scale datasets. The authors extend this approach to contrastive learning, demonstrating its effectiveness in handling high-dimensional embeddings.
2. The paper addresses the challenge of efficiently sampling hard negative examples in contrastive learning. Previous research has shown that hard negative samples significantly enhance the performance of contrastive learning. The proposed method not only maintains the quality of hard negative samples but also reduces the computational overhead associated with traditional methods.
3. The paper includes a theoretical analysis of the LSH method, providing insights into its behavior and effectiveness. This complements prior theoretical work on LSH and its applications in various fields.

**Theoretical Claims:**

1. The paper introduces LSH by mapping high-dimensional vectors to low-dimensional binary spaces, thereby reducing computational complexity while maintaining similarity. It also includes random rotation matrices and centralization, ensuring that the hash function can evenly distribute data points, thereby increasing the probability of hash collisions. This is theoretically sound.
2. The paper proposes the application of LSH to hard negative sampling in contrastive learning. Through LSH, it is possible to efficiently find negative samples similar to the anchor, thereby improving learning performance. Experimental results on different datasets support the effectiveness of LSH in contrastive learning. It has been proven that the LSH method maintains computational efficiency without significantly reducing performance.

---

> ### Author Rebuttal · Authors · 2025-03-31
>
> We thank the reviewer for their comments, and begin by addressing them below:
> 1. We agree with the reviewer that LSH has been extensively studied in the context of approximate nearest neighbor search. However, its application in contrastive learning, especially for hard negative sampling during training, remains underexplored. Moreover, in the few cases where it has been used, its application is often limited to specific modalities.
> For example, Gillick et al. (Learning Dense Representations for Entity Retrieval 2020) employed quantization-based fast inner product search (Quantization based Fast Inner Product Search Guo et al., 2015) for entity retrieval. While effective, this method relies on data-dependent codebook training, introducing additional complexity and limiting generalizability.
> In contrast, our approach uses data-independent LSH to efficiently sample hard negatives in a way that is broadly applicable across tasks and modalities. Unlike data-dependent methods, our projection does not require continuous retraining, which is a common limitation as embeddings evolve during training. We will make this distinction more explicit in the Research Gap and Introduction. As noted in our response to reviewer SDTH, we have also quantified the benefits of our method in terms of computational savings per epoch, demonstrating its practical scalability in large-scale settings.
>
> 2. Thanks for the suggestion. Modality-specific LSH methods such as bag-of-words approaches (BM25) for hard negative search have been explored (Karpukhin et al. 2020, Dense Passage Retrieval for Open-Domain Question Answering). However, they can also be prone to false negatives, and perform worse than embedding-based approaches (Xiong et al. 2021, Approximate Nearest Neighbor Negative Contrastive Learning for Dense Text Retrieval). In contrast, our approach is modality-agnostic as we operate directly on embeddings and generalize across text, image, and multimodal data. This allows for a unified and scalable retrieval framework without relying on modality-specific heuristics.
> 3. We based our LSH sampling approach mostly on Wang et al. 2015 (Learning to Hash for Indexing Big Data - A Survey) who refer to the well known fact that the collision probability of two points is equal to their similarity but to our understanding do not offer deeper insights regarding the similarity between the point with the nearest computed hamming distance and the anchor/query. Other LSH related work such as Har-Peled et al. 2012 (Approximate Nearest Neighbor: Towards
> Removing the Curse of Dimensionality) states that for a defined radius, and a nearest neighbor existing in that radius of a query point, we find an approximate neighbor in some $cr$ neighborhood of the query point. We on the other side always return the neighbor with the smallest projected hamming distance to the query/anchor point which is different.
> Therefore we can not build upon their work and offer the following insight instead. As the hamming distance is the sum of d independent Bernoulli trials it corresponds to a binomial distribution with success rate equal to their cosine distance. For large bit sizes this is approximating a normal distribution with mean equal to bit size times cosine distance. When a point x is closer to the anchor than a point y this should ideally reflect in a smaller hamming distance, i.e. $\mathrm{HammDist}(c, y) - \mathrm{HammDist}(c, x) > 0$. Let $Z=\mathrm{HammDist}(c, y) - \mathrm{HammDist}(c, x)$. As the difference of two normal distributions $Z$ is again a normal distribution with mean $\mu_Z=n (\mathrm{sim}(c,y)- \mathrm{sim}(c,x))$ and variance $\sigma_{Z}^2 = n \left( (1 - \mathrm{sim}(c,x))  \mathrm{sim}(c,x) + (1 - \mathrm{sim}(c,y))  \mathrm{sim}(c,y) \right)$. Therefore $$P(Z<=0)=P\left(\frac{Z-n (\mathrm{sim}(c,y) - \mathrm{sim}(c,x)) )}{ \sqrt{(n ((1- \mathrm{sim}(c,x)) \mathrm{sim}(c,x) + (1- \mathrm{sim}(c,y)) \mathrm{sim}(c,y))}}\right).$$ with increasing bit size the probability of $\mathrm{HammDist}(c,y) < \mathrm{HammDist}(c,x)$ decreases. We already outlined this in L194-202 but we will strengthen this section with the above comprehensive insight. Thank you for the suggestion!
> 4. Thanks for the suggestion. We use a random hyperplane-based LSH, chosen for its simplicity, strong performance on high-dimensional embeddings, and efficient GPU implementation.
> While more advanced variants such as cross-polytope LSH offer improved theoretical recall, they involve more complex encoding and decoding schemes that are less suitable for large-scale, batched GPU workflows. Similarly, multi-probe techniques are complementary to our approach and could be integrated to further improve recall, but we found our current setup sufficient for effective hard negative mining. We will add this in our Discussion to address this more clearly.
>
> We also thank the reviewer for pointing out the citation unification, we will enhance this further!

---

> > ### Comment · Reviewer_udPs · 2025-04-04
> >
> > The authors addressed the core issues raised in the review by clarifying the versatility of their data-agnostic LSH method and strengthening the theoretical link between Hamming distance and cosine similarity through probabilistic modeling. However, critical gaps remain unresolved. Their application of LSH still falls under existing technical adaptations, lacking substantial justification for its inherent innovation. Additionally, while the binomial approximation of Hamming distance is theoretically reasonable, it overlooks practical impacts of high-dimensional sparsity (e.g., long-tail effects) on distribution. Although the method claims "modality-agnostic" applicability, its generality is weakened by the absence of validation on more complex multimodal tasks. Consequently, I keep my original score.

---

### Official Review · Reviewer_SDTH · 2025-03-17

**Overall Recommendation:** 3

**Summary:**

This paper proposes to use locality-sensitive hashing to extract nearest neighbors as hard negatives when performing contrastive learning to train sentence embedding. The authors perform experiments to verify that the proposed method can achieve almost the same embedding quality and that the search time runs an order faster than the most accurate baseline.

Update after rebuttal:

The authors' response addressed my concern regarding the motivation of the new method. My score remains unchanged.

**Claims And Evidence:**

yes

**Essential References Not Discussed:**

yes

**Experimental Designs Or Analyses:**

yes

**Methods And Evaluation Criteria:**

yes

**Other Comments Or Suggestions:**

n/a

**Other Strengths And Weaknesses:**

Strength:
1. The proposed method is clean and effective.
2. The authors use math formulas and visualization to explain how LSH is used in a clear way
3. Comprehensive experiments were done on the relationship between the number of bits used in LSH and the hard negative quality. And how hard negative quality affects final sentence embedding quality.

Weakness:
1. Perhaps the authors can try stronger datasets where the gaps between different methods are more obvious.

**Questions For Authors:**

1. I am curious how much is the hard negative search time compared to the training time. If most of the time is spent on training the models on the samples within epochs, then optimizing the search time seems to be not well motivated.

**Relation To Broader Scientific Literature:**

n/a

**Theoretical Claims:**

n/a

---

> ### Author Rebuttal · Authors · 2025-03-31
>
> We thank the reviewer for his insightful comments. To address the concern about our motivation, we measured the training times per epoch (averaged over 3 epochs) using a DGX-2 on different datasets and compared these to the pre-epoch HN search times shown in Figure 1 and the search time with 128 bit:
>
> | Dataset   	| Dataset size - train | Training time per epoch without search time (s) | Search time per epoch LSH 128 bit (s) | Search time per epoch Pre-Epoch HN (s) | Search time proportion per epoch when using Pre-Epoch HN |
> |---------------|--------------|---------------------------------------------|----------------------------------------|------------------------------------------|----------------------------------|
> | CVUSA/CVACT   | 35,532   	| 181.91                                 	| 0.35                            	| 25.10                              	| 12.12%                       	|
> | VIGOR     	| 40,007   	| 395.20                                  	| 0.437                              	| 41.10                               	| 9.42%                        	|
> | SOP       	| 59,551   	| 137.24                                 	| 0.717                              	| 69.04                               	| 33.46%                       	|
> | InShop    	| 25,882   	| 65.57                                   	| 0.21                            	| 13.33                               	| 16.89%                       	|
> | MS Marco  	| 532,736  	| 566.92                                  	| 33.57                              	| 7553.27                              	| 93.01%                       	|
>
> We agree this provides important details to our motivation, and we will include this table in the supplementary materials. We thank the reviewer for encouraging us to clarify this aspect further.
>
> We appreciate the reviewer's comment regarding the choice of dataset. As stated in our research gap (Section 2.3), our goal is to evaluate whether a binarized, low-dimensional representation can match the performance of full-scale embeddings in the context of hard negative sampling for contrastive learning. The results in Tables 1 and 2 indicate that the performance gap remains small, which is consistent with our research goal. If there were a significant performance drop when comparing pre-epoch incremental sampling to our LSH-based sampling, it would suggest that LSH is ineffective for efficient hard negative selection. However, the minimal performance differences observed support our hypothesis and demonstrate that LSH provides a practical balance between computational efficiency and representational effectiveness.

---

### Decision · Program_Chairs · 2025-05-01

**Decision:**

Reject

**Comment:**

This paper proposes a simple yet practical method that applies LSH to efficiently sample hard negatives in contrastive learning. The reviewers appreciated its computational efficiency, modality-agnostic design, and empirical validation across multiple datasets. However, concerns persist about the lack of theoretical grounding, limited novelty over existing LSH-based approximate nearest neighbor methods, and insufficient comparisons to alternative dimensionality reduction techniques. The authors provided detailed rebuttals, clarifying implementation choices and quantifying training efficiency gains, but there remain concerns about the paper’s originality and broader impact. Given the empirical results but concerns over innovation and rigor, I recommend a rejection.